# Exploring the Optimal Operating MR Contrast for Brain Ventricle Parcellation

**Savannah P. Hays**[1]                                                SHAYS6@JHU.EDU

[1] *Department of Electrical and Computer Engineering, Johns Hopkins University, USA*

**Lianrui Zuo**[1,2]                                                LR_ZUO@JHU.EDU

[2] *Laboratory of Behavioral Neuroscience, National Institute on Aging, National Institutes of Health, USA*

**Yuli Wang**[3]                                                YWANG687@JHMI.EDU

[3] *Department of Biomedical Engineering, Johns Hopkins School of Medicine, USA*

**Mark G. Luciano**[4]                                                MARKLUCIANO@JHU.EDU

[4] *Department of Neurosurgery, Johns Hopkins School of Medicine, USA*

**Aaron Carass**[1]                                                AARON_CARASS@JHU.EDU

**Jerry L. Prince**[1]                                                PRINCE@JHU.EDU

## Abstract

Recent development in magnetic resonance (MR) harmonization has facilitated the synthesis of varying MR image contrasts while preserving the underlying anatomical structures. This enables an investigation into the impact of different T1-weighted (T1-w) MR image contrasts on the performance of deep learning-based algorithms, allowing the identification of optimal MR image contrasts for pretrained algorithms. In this study, we employ image harmonization to examine the influence of diverse T1-w MR image contrasts on the state-of-the-art ventricle parcellation algorithm, VParNet. Our results reveal the existence of an optimal operating contrast (OOC) for VParNet ventricle parcellation, achieved by synthesizing T1-w MR images with a range of contrasts. The OOC for VParNet is not of the same MR image contrast of any of the training data. Experiments conducted on healthy subjects and post-surgical NPH patients demonstrate that adjusting the MR image contrast to the OOC significantly enhances the performance of a pretrained VParNet, thereby improving its clinical applicability.

**Keywords:** MR imaging, Harmonization, Ventricles

## 1. Introduction

VParNet (Shao et al., 2018, 2019) is a deep learning based ventricle segmentation algorithm, demonstrating state-of-the-art performance on both healthy and normal pressure hydrocephalus (NPH) subjects. However, the development and evaluations of VParNet has only focused on a healthy (Marcus et al., 2007) and an NPH (Shao et al., 2019) cohort. This limits its application in clinics due to the inherent variability in magnetic resonance (MR) image contrast. As we show in Sec. 3, VParNet can fail on images acquired from cohorts beyond its training scope, highlighting the challenges of domain shift. In this paper, we propose a framework to improve the applicability and accuracy of VParNet using MR harmonization without retraining the parcellation model. Our framework is based on a recently proposed MR harmonization method (Zuo et al., 2021a,b), which synthesizes MR images with different

Figure 1: DSC based heatmap showing the OOCs $A$ and $B$ of VParNet identified by a grid search of the CALAMITI contrast space. The gray area in the upper-right hand corner, is a region of CALAMITI contrast space associated with $T_2$-weighted MRIs and was not explored in our grid search of $T_1$-weighted OOCs for VParNet.

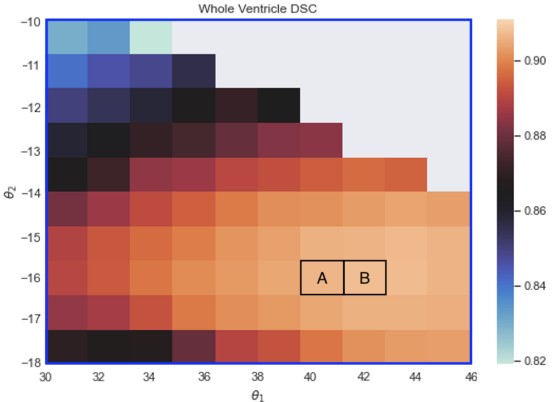

contrasts while preserving the underlying anatomy. Previous work by Hays et al. (Hays et al., 2022) has demonstrated the potential use of harmonization to evaluate the impact of $T_1$-w MR image contrast on whole brain segmentation. Following (Hays et al., 2022), we first use image harmonization to quantitatively estimate the operating contrasts of VParNet. Furthermore, we demonstrate there exists an *optimal* operating contrast (OOC) for VParNet. After adjusting image contrast to the identified OOC, we show that the performance of a pretrained VParNet can be further boosted even on the same healthy cohort it was trained on. Experiments on NPH cohorts with diverse MR image contrasts demonstrate the generalizability of VParNet's performance when using the OOC.

## 2. Methods

Using CALAMITI (Zuo et al., 2021a,b), we are able to generate synthetic MR images with the same underlying anatomy, $\boldsymbol{\beta}$, and an arbitrary contrast, $\boldsymbol{\theta}$. Thirty-five $T_1$-w MR images of healthy subjects were first encoded into the CALAMITI contrast and anatomy space. These 35 images (which were not used in training VParNet) were acquired on a 1.5T Siemens scanner, and manually delineated (including the ventricles) by experts from Neuromorphometrics Inc. (NMM) (Marcus et al., 2007). We observe that all of these images have MR image contrast within the broad range of CALAMITI's training data. In order to find the OOC for VParNet, we considered the MR image contrasts within CALAMITI training data, which is represented by the blue rectangle shown in Fig. 1. Dividing this rectangle into a $10 \times 10$ grid, we rejected 21 locations with non-$T_1$-w MR image contrasts, leaving 79 candidates for the OOC. We then combined each candidate (target) contrast with the encoded anatomical representations ($\boldsymbol{\beta}$-values) of eight NMM subjects to generate synthetic MR images. In total there are 632 synthetic images (8 subjects times 79 candidate contrasts) generated by CALAMITI. These synthetic images and the corresponding ventricle delineations were used to quantitatively evaluate the performance of VParNet on the 79 candidate contrasts. The average Dice similarity coefficient (DSC) of VParNet on the eight sample subjects of each candidate contrast is shown in Fig. 1. We identified the OOCs of VParNet (labeled as $A$ and $B$) by selecting the candidate contrast with the highest mean DSC. Interestingly, the OOCs represent different MR image contrasts than VParNet training data in CALAMITI contrast space.

Figure 2: Example images PS-NPH subjects. The original PS-NPH images have a surgical artifact that is mislabeled as part of the ventricle system by VParNet, while the images harmonized to an OOC have no such mislabeling.

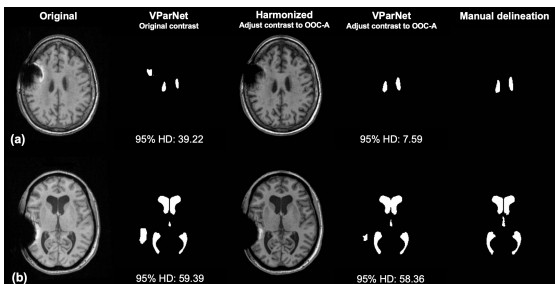

## 3. Experiments and Results

We adjust the contrast of the remaining 35 subjects from the NMM dataset that were not included in VParNet training to the two selected OOCs. There was a significant ($p < 0.01$) improvement between VParNet performance on the original and harmonized images for the 4th ventricle, left lateral ventricle, right lateral ventricle, and whole ventricle. The VParNet DSC for the whole ventricle on the original MR images was $0.8859\pm0.039$. After harmonization, the DSC improved to $0.8932\pm0.031$ and $0.8947\pm0.032$ using OOCs $A$ and $B$, respectively. This result demonstrates the ability of the OOC to enhance VParNet's performance even on the original training cohort. This boost motivated us to explore the impact of image harmonization on datasets beyond VParNet's training scope. Our post-surgery NPH (PS-NPH) dataset consists of six patients with corresponding manual delineations. These patients underwent brain surgery to remove excess CSF from their ventricles. PS-NPH MR images often contain artifacts hindering the performance of automatic parcellation methods. In many cases, these artifacts led to mislabelling during automated image processing. The $T_1$-w MR images of the six patients were acquired at different clinical centers, distinct from the cohorts where VParNet was trained. We used image harmonization to adjust the contrast of the PS-NPH $T_1$-w MR images to the previously identified OOCs (Targets $A$ and $B$) and observed improved performance of VParNet. Figure 2 shows the results for two post-surgery NPH patients. VParNet's performance on the original images is considerably impacted by the diverse MR image contrast and surgery artifacts, causing the artifact to be mislabeled as part of the ventricular system. After adjusting the images to the OOCs, VParNet shows improved performance. The 95% HD improves from 39.22 to 7.59 for subject (a) in Figure 2. The improvement for subject (b) is more subtle.

## 4. Discussion and Conclusion

In this paper, we have demonstrated the improved performance of VParNet for both healthy and NPH patients by adjusting the input image contrast using harmonization, without retraining VParNet. We successfully identified two OOCs of a pretrained VParNet with a grid search in CALAMITI contrast space. After contrast adjustment, we demonstrate improved VParNet performance on data from the same cohort as VParNet training and data from different cohorts. For the PS-NPH subjects, VParNet showed improved DSC and 95% HD after contrast adjustment; however, mislabeling due to the post-surgery artifacts persists. The OOCs $A$ and $B$ are specifically for the whole ventricle label and may differ if the evaluation focuses on a particular ventricle.

## Acknowledgments

This work was supported in part by the NIH / NINDS under grant U01-NS122764 (PI: M.G. Luciano) and in part by the Intramural Research Program of the NIH, National Institute on Aging. Portions of the used data in this study was conducted retrospectively using human subject data made available by Neuromorphometrics Inc. Ethical approval was not required as confirmed by the license attached with this data. The remainder of the data was acquired in line with the principles of the Declaration of Helsinki. Approval was granted by an IRB Committee of the Johns Hopkins School of Medicine with approval ID IRB00305245 (approved January 13, 2022).

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
