# OpenReview forum: "Exploring the Optimal Operating MR Contrast for Brain Ventricle Parcellation"
_MIDL.io/2023/Short_Paper_Track — MIDL 2023 Short paper track Poster_

### Official Review · Reviewer_Pfvq · 2023-04-10
**Simple but intriguing result**

**Rating:** 7
**Confidence:** 5

**Review:**

In this paper, the authors assess how tinkering with the contrast of an MRI scan using an unsupervised harmonization technique affects the performance of a segmentation neural network with fixed weights. Very intriguingly, the paper also shows that this contrast is *not* the contrast of the source domain. The paper does not go much deeper into why this may be the case (the space constraints are severe, to be fair)  but I think that it could lead to interesting discussion during the conference.

PS: a (very minor) slap on the wrist to the authors, for withholding their names & institutions on the pdf, which saves space (maybe they were not aware that short papers were single-blind, in which case, the slap on the wrist is for not reading the instructions!)

---

### Official Review · Reviewer_ymv4 · 2023-04-24

**Rating:** 7
**Confidence:** 5

**Review:**

This paper presents a way of improving VParNet for brain ventricle parcellation without retraining by incorporating CALAMITI  and defining optimal operating contrast (OOC) for image harmonization.

It is a straightforward integration of data harmonization and deep learning methods, which shows outperformance over the original VParNet.